# Treatment of Membrane Cleaning Wastewater from Thermal Power Plant Using Membrane Bioreactor

**DOI:** 10.3390/membranes12080755

**Published:** 2022-07-31

**Authors:** Wenxiu Zhang, Xiaoyi Xu, Guanghui Zhang, Shengjiang Jin, Lihua Dong, Ping Gu

**Affiliations:** School of Environmental Science and Engineering, Tianjin University, Tianjin 300350, China; misszhangwx@163.com (W.Z.); 2021214109@tju.edu.cn (X.X.); 2015214083@tju.edu.cn (S.J.); lihua.dong@tju.edu.cn (L.D.); guping@tju.edu.cn (P.G.)

**Keywords:** membrane bioreactor, membrane cleaning wastewater, hydraulic residence time, solid residence time, membrane fouling

## Abstract

An integrated membrane bioreactor (MBR) with synthetic RO membrane cleaning wastewater from a thermal power plant was used to study the long-term operating characteristics, membrane fouling, and cleaning of membrane fouling. The results show that the MBR had a great removal effect on mainly an organic pollutant (citric acid) with an average of 98.4% rejection, and the concentration of organics in the effluent also achieved “Discharge standard of pollutants for municipal wastewater treatment plant” (GB12/599-2015). The optimal operating conditions were as follows: the membrane flux was 8 L/(m^2^·h); the hydraulic retention time (HRT) was 4 h; the sludge retention time (SRT) was 15 d, and the pH value was 6~7. A membrane fouling analysis showed that the resistance of the cake layer and the concentration polarization were the main components of membrane fouling. When the specific flux (*SF*) decreased to 10 L/(h·m^2^ mH_2_O), the membrane module was cleaned by tap water and then soaked in 0.05 wt% hydrochloric acid (HCl) and 3000 mg/L sodium hypochlorite (NaOCl) for 1 h and 3 h, respectively. Finally, the membrane flux could be recovered to 84.9% compared to the new membrane.

## 1. Introduction

While thermal power plants provide the public and industrial users with abundant electrical energy, a large amount of circulating cooling wastewater and membrane cleaning wastewater is discharged [1,2]. Raw water cannot be used directly in a circulating system or for high-pressure boilers due to the total hardness and presence of higher concentrations of total dissolved solids and other salts and chlorides, which can strongly corrode and scale in boiler tubes and must be removed prior to being fed into the boilers and to create a barrier to heat transfer resulting in higher fuel consumption and maintenance costs [3]. RO is a commonly used process to obtain demineralized water for high-pressure boiler operation in thermal power plants [4,5]. RO membrane pores are blocked by the inorganic scales and organic compounds step by step in the operation, which will cause the fouling of membranes [6,7]. In order to remove the foulants and recover the membrane flux, physical and chemical cleaning are suggested [8,9]. After the fouled RO membranes are cleaned using chemical agents, membrane cleaning wastewater, which contains large quantities of acid or alkali [10,11], will be produced that need to be safely treated to prevent them from threatening the water environment.

For inorganic pollutants, acids, such as hydrogen chloride (HCl), oxalic acid, and citric acid, have been proven to achieve effective removing [12]. Citric acid possesses a strong buffering capacity and is useful for cleaning membranes prone to inorganic fouling making it a better alternative than mineral acids, which may cause pH damage to the membranes [13,14,15,16]. As a result, the membrane cleaning wastewater produced from the thermal power plant usually contains large amounts of citric acid. Pan et al. used a biologically active filter (BAF)/O_3_ process for the advanced treatment of citric acid wastewater, and the average removal efficiency of COD, chromaticity, and UV254 was 76%, 93%, and 27%, respectively, and the effluent could be used for production processes of citric acid [17]. Li et al. used UASB-contact oxidation-air support to treat citric acid wastewater. The operation shows that under the influent condition of SS = 3427 mg/L, COD = 18,853 mg/L, and BOD_5_ = 11,778 mg/L, SS, COD, and BOD_5_ of the effluent were 147.5 mg/L, 223 mg/L, and 51.9 mg/L [18].

A membrane bioreactor (MBR) combines both biological treatment and physical separation to treat domestic and industrial wastewater, which replaces the secondary sedimentation with membrane modules in the traditional treatment methods [19]. The working principle is to firstly use the activated sludge in the reactor to degrade the biodegradable organic matter in the sewage, and the membrane module is to separate the microorganisms and some macromolecules. MBRs have obvious advantages that many other biological treatment processes cannot match [20]: 1. The device is more compact and has a small footprint; 2. The effluent water quality is excellent and stable; 3. The system has high nitrification efficiency and a high sludge concentration (MLSS); 4. The microorganisms are completely trapped in the reactor, and the complete separation of the reactor hydraulic retention time (HRT) and sludge retention time (SRT) is achieved, which makes the operation control more flexible and stable; 5. The reactor has high mass transfer efficiency; 6. It is easy to manage. Since the 21st century, MBR has been widely used in wastewater treatment [21,22,23,24,25] and is attributed to the reduction of the membrane material. However, the major drawback of MBR technology is membrane fouling, which affects the operating flux and the life of membranes [26,27]. Fouling in MBRs is primarily caused by microbial deposition/growth and microbial product accumulation on membranes. Characteristics of microorganisms and microbial products in MBRs strongly depend on operating conditions of MBRs [28], including operating mode, rate of aeration, solid retention time (SRT), hydraulic retention time (HRT), food–microorganism (F/M) ratio, organic loading rate (OLR), chemical oxygen demand/nitrogen (COD/N) ratio, and temperature [27].

Membrane cleaning can effectively control membrane fouling including physical cleaning and chemical cleaning. A combination of physical cleaning and chemical cleaning is common in the actual membrane cleaning operation [29]. Physical cleaning mainly removes reversible contaminants in the membrane surface or membrane pores, and the methods mainly include aeration, backwashing, ultrasonication, sponge scrubbing, and water washing. Chemical cleaning can partly remove nonreversible fouling, and commonly used chemical agents include alkali cleaning agents, acid cleaning agents, oxidizing cleaning agents, and surfactants, which can greatly restore membrane flux, but the cleaning waste can sometimes cause secondary fouling and increase membrane maintenance and operation costs [30].

In the paper, the treatment of the membrane cleaning wastewater produced from the RO unit of the circulating cooling water system in a thermal power plant was investigated. An integrated-type (also called submerged type) MBR was used for the treatment of the membrane cleaning waste, and the effect of operation parameters and conditions on the removal of organic compounds and membrane fouling were explored.

## 2. Materials and Methods

### 2.1. Raw Water and Chemical Agents

Water quality analysis of the RO membrane cleaning wastewater from a thermal power plant was conducted, and it was determined that the main component of the wastewater was citric acid. According to the measurement results, the target parameters of the test raw water were: COD = 620 mg/L, COD:TN:TP = 100:5:1 (to provide carbon, nitrogen, and phosphorus required by microorganisms). The specific dosage of citric acid 0.88 mg/L, ammonium chloride 0.12 mg/L, potassium dihydrogen phosphate 0.03 mg/L, and sodium hydroxide 0.39 mg/L and the pH value of the raw water were controlled at 6~7. The test raw water was prepared with tap water.

Oher chemicals used in the study including sodium hypochlorite (NaClO, 3000 mg/L), sodium hydroxide (NaOH, 0.2 wt%), and hydrochloric acid (HCl, 0.05 wt%) were all analytical reagents (ARs) and purchased from Tianjin Guangfu Fine Chemical Research Institute (Tianjin, China).

### 2.2. Membrane Bioreactor

The membrane bioreactor was made of organic glass, and the inner diameter and height were 127 mm and 1.5 m, respectively. There was a feeding inlet on the top and a sludge discharge valve at the bottom of the reactor to discharge excess sludge. A perforated air diffuser was placed at the bottom and provided the dissolved oxygen in the wastewater to alleviate membrane fouling. The microfiltration membrane located above the diffuser was manufactured by Tianjin Maotian Membrane Technology Co., Ltd. Tianjing, China, detail in Table 1. The treatment capacity of MBR was 4 L/h.

The working procedure of MBR is as follows: the simulated wastewater was lifted from the tank to the reactor by a pipeline pump. The effluent was intermittently drawn by a peristaltic pump through the microfiltration membrane module with an operation mode of suction 8 min and idle 2 min in every 10-minute cycle. The blower continuously aerated and supplied the air through the perforated aeration diffuser to provide the oxygen required for the metabolism of microorganisms, and at the same time, it can also play a role in alleviating membrane fouling. The volume ratio of supplied air to treated wastewater was 60:1. HRT was designed as 4 or 6 h with a SRT of 30 d, and HRT of 2 or 4 h with a SRT of 15 d. The parameters representing the wastewater quality including organic compounds, sludge performance, pH values, and turbidity were investigated in the test.

### 2.3. Sludge Acclimation in Membrane Bioreactor

Before the test, the activated sludge taken from the secondary sedimentation tank of the municipal wastewater treatment plant was used for inoculation and acclimation. The inoculated sludge and 10 L of raw water were added to the reactor. The initial concentration of the sludge was 1.2 g/L. After aeration for 24 h, the mixed liquor in the reactor was settled for 1 h, and 1/3 of the supernatant was discharged, and the same amount of raw water was added. The operation was repeated for three days. On the 4th day, raw water was added with a cycle of 12 h, and the acclimation continued for a week. When COD in the supernatant was below 30 mg/L, it indicates that the acclimation was completed, that is, the reactor is considered to be able to operate stably.

### 2.4. Membrane Fouling and Cleaning

Membrane specific flux (*SF*) is an important parameter index to evaluate membrane fouling. It is defined as the amount of water passing through the unit membrane filtration area in unit time under unit pressure difference across the membrane. The calculation formula is as Equation (1). The membrane fouling rate is calculated by Equation (2).
(1)$$SF=\frac{Q}{A\cdot TMP}$$
(2)$$\sigma =\frac{\Delta SF}{\Delta t}$$
where *SF* is the specific flux, L/(h·m^2^·mH_2_O); TMP is the pressure difference across membrane using water head, m; *Q* is the flowrate, L/h; *A* is the filtration area, m^2^; σ is the fouling rate, 1/h^2^; Δ*SF* is the difference of SF at different operation times, L/(h·m^2^·mH_2_O); Δ*t* is the difference of the operation time, h.

The membrane was fouled along with operation, and the *SF* decreased step by step. When *SF* decreased to 10 L/(h·m^2^·mH_2_O), the fouled membrane should be cleaned to recover. *S**F* was measured once spending 30 min in the acid and sodium hypochlorite cleaning. Physical and chemical cleaning were both used in the test to remove the foulants in the fouled membrane when the module in MBR was blocked during operation for some time. Tap water, acid, alkali, and sodium hypochlorite were used in the cleaning:(1)Water cleaning: use tap water to flush the sludge on the surface of the module and then soak it in tap water for 120 min and use tap water to flush again;(2)Acid cleaning: soak the membrane in 0.05 wt% hydrochloric acid for 60 min and then wash it with tap water until neutral;(3)Sodium hypochlorite cleaning: soak the membrane in sodium hypochlorite of 3000 mg/L for 30–180 min and then wash it with tap water until neutral.

### 2.5. Analytical Methods

COD was determined by the titration using potassium bichromate. pH and dissolved oxygen were analyzed by a multiparameter meter (Model HACH HQ40D, HACH, Loveland, CO, USA). Anions were made on an ion chromatograph (Thermo ICS-1100, Thermo Fisher Scientific, Waltham, MA, USA). Turbidity measurements were taken on a photoelectric turbidimeter (Model HACH 2100P, HACH, Loveland, CO, USA). MLSS was measured by weight method.

## 3. Results

### 3.1. Effect of SRT and HRT on Removal of COD

Figure 1 shows the contents and rejections of COD in MBR influent and effluent. It can be seen that when the COD in MBR influent was 500–700 mg/L, the COD removal efficiency under different operation conditions was more than 96%, and the COD in effluent was less than 20 mg/L, which can meet the requirements of the wastewater discharge standard.

The average COD removal efficiency was 97.6% when SRT was 30 d and HRT was 6 h, while HRT was 4 h for 97.0%, indicating that the COD capacity in the MBR was sufficiently removed by active sludge with 4 h. We shortened SRT to 15 d and HRT to 4 h and 2 h, respectively, and continued to study the effects of SRT and HRT on the removal of COD. As can be seen from Figure 1, once HRT was reduced to 2 h, the effluent COD rapidly rose to more than 100 mg/L, and the COD removal efficiency fluctuated between 65% and 84%. The result shows that the MBR removal effect was significantly worse when HRT is 2 h, and HRT should be required to be at least 4 h. This is because HRT was too short to degrade COD by active sludge, that is, the sludge load exceeded the capacity of MBR. Therefore, it can be concluded that 4 h was the inferior limit of HRT, which could ensure MBR operates normally.

The effect of SRT on the removal of COD from MBR can also be seen from Figure 1. Under the same conditions of HRT (4 h), when the SRT was set to 15 d and 30 d, the average COD removal efficiency was 98.4% and 97.0%, and the average COD concentration of effluent was 9.5 mg/L and 18.7 mg/L, respectively. Obviously, the shorter the SRT, 15 d, the better the effluent quality was. In the case of HRT, reducing SRT could speed up the sludge renewal rate to keep the activity of the sludge in MBR, which is more favorable for the degradation of COD. However, the smaller SRT, the greater the amount of sludge, which is detrimental to the cost of subsequent sludge treatment. Therefore, taking into account the removal of COD and the cost of subsequent sludge treatment, the optimal HRT and SRT were determined to be 4 h and 30 d, respectively.

### 3.2. Effect of SRT and HRT on Sludge Performance

Figure 2 shows the variations of the sludge volume index (SVI) in the test, which was quite different from Figure 1. The SVI curve had a steady increase in the first stage (SRT = 30 d, HRT = 6 h) and early second stage (SRT = 30 d, HRT = 4 h) and then decreased in the later second stage. The SVI varies greatly in the first and second phases. It was basically stable in the third stage (SRT = 15 d, HRT = 4 h). Then, it decreased quickly in the fourth stage (SRT = 15 d, HRT = 2 h). An SVI below 200 mL/g indicates good settling characteristics of active sludge. In this test, the activated sludge of MBR showed good settleability.

The SVI showed a linear increase from 49 to 167 mL/g. The reason for this was that at the beginning of the MBR operation, the sludge concentration was at a low level, and the influent provided enough organic matter for microorganisms to grow and reproduce, and as the running time increased, the sludge concentration also grew rapidly. However, with SRT of 30 d, the SVI then decreased to 79 mL/g. The fluctuation of SVI was attributed to long SRT, in which the sludge was less active in MBR than that in the aeration tank in the conventional wastewater treatment plant (WWTP).

When SRT was reduced to 15 d, it became stable with the corresponding HRT of 4 h in which the operation parameters were approximately the same as the WWTP. When HRT was further reduced to 2 h, the contact between the activated sludge and the organic compounds in MBR was greatly decreased, which led to a reduction of the biodegradation probability of the organic compounds. In addition, the short HRT would cause activated sludge to not receive enough food to maintain the metabolism, and the stability of the activated sludge was inhibited, hence SVI started to decrease in the final stage of the operation. It could be concluded that the HRT close to WWTP could make the activated sludge stable and more or less had an influence on the sludge performance. In this section, the recommended SRT and HRT were 15 d and 4 h, respectively.

### 3.3. Effect of SRT and HRT on pH Values and Turbidity

The variations of the pH values in the influent and effluent with different SRT and HRT are shown in Figure 3. All of the pH values were generally stable in the four stages of the operation. The pH values in the influent ranged from 6.06 to 6.76, and the corresponding pH values in the effluent ranged from 8.09 to 8.69.

The pH values in the effluent increased approximate two units after wastewater treatment, attributed to biodegradation of the citric acid in the membrane cleaning wastewater. The pH value difference between the influent and effluent was also relatively stable, and it implied that the removal of the citric acid in wastewater did not fluctuate obviously. The differences of the pH values per unit COD (ΔpH/ΔCOD) were also investigated, as shown in Figure 4. The trend of the variations of ΔpH/ΔCOD were highly consistent with that of ΔpH. It could be calculated that the difference of pH values would increase 3.25 × 10^−3^ L/mg when unit COD was removed from the wastewater in the stable operation.

Figure 5 provides the turbidity in the effluent of MBR. All of the turbidities were less than 0.7 NTU, which means that the microfiltration module had a very strong separation on suspended solids in the mixed liquor of the reactor. Only 13% of the effluent turbidity was beyond 0.4 NTU, and 87% was less than 0.4 NTU. A total of 42% of the effluent turbidity ranged from 0.1 to 0.2 NTU, nearly a half of all the turbidity. A total of 27 and 18% of the turbidity was distributed in the ranges of 0.2–0.3 and 0.3–0.4, respectively. Even if the removal of COD greatly decreased in the final stage, the effluent turbidity was still stable; as a result, the effluent turbidity was not greatly affected by the operation parameters.

### 3.4. Effect of SRT and HRT on Membrane Fouling

A promising solution for membrane fouling reduction in membrane bioreactors (MBRs) could be the adjustment of operating parameters of the MBR.

Membrane fouling was caused by several factors including the membrane material, wastewater quality, operation parameters, mixed liquor, and so on [31]. The effect of SRT and HRT on membrane fouling rates was discussed in this section.

Figure 6 shows the variations of *SF* in the operation with the different SRT and HRT, which could be used to investigated membrane fouling rates in MBR. *SF* could be calculated using Equation (1). Variations of *SF* followed a similar trend that *SF* quickly decreased in the initial part and then gradually decreased with a stable decreasing rate in each stage, that is to say, the microfiltration membrane module had been seriously fouled at first, which was attributed to the formation and growth of the cake layer on the membrane surface. The cake layer attached to the membrane surface grew step by step, which was simultaneously erased from the membrane surface by aeration [32,33]. When the attachment on and erosion from the membrane surface reached the dynamic balance, the thickness of the cake layer would not greatly increase with the operation. The procedure could be explained using the separation mechanism of the microfiltration that included sieve, adsorption, and bridging [34]. At first, the activated sludge in the mixed liquor were cumulated near the membrane, blocked the pore, and attached on the surface, which reduced the effective filtration area and led to a fast fouling; therefore, three mechanisms were contributed to the initial fouling, and the fouling rate was the largest. After the cake layer formed, most of the contaminants were retained by the cake layer, and only small-sized particles could pass through and enter the membrane; thus adsorption and bridging were responsible for the stable fouling.

In Figure 6, the slope of the fitting line of *SF* could be treated as the membrane fouling rate. If the fitting lines were plotted for the whole and stable part of the curve of *SF*, the gross, fast, and stable fouling rates in each stage could be calculated using Equation (2), as shown in Table 2. The removal of COD were much less in the third stage than others in which the biodegradation of the organic compounds was greatly decreased; therefore, the membrane filtration was highly loaded and contributed to the serious fouling in the third stage. When the cake layer on the membrane surface was formed, there was a 75% reduction of the fouling rates from the fast to the stable part in the second stage. The stable fouling rate was a quarter of the fast-fouling rate and 40 percent of the gross values, respectively.

In a word, SRT and HRT had a great effect on the membrane fouling rate. When SRT was reduced from 30 d to 15 d and HRT from 4 h to 2 h, respectively, a great difference in fouling rates was observed in the operation, especially the fast fouling in Stages Ⅱ and Ⅲ. As a consequence, the fouling was dependent on both the SRT and HRT, and the larger the SRT and HRT were, the smaller the fouling rate was.

### 3.5. Membrane Cleaning and Flux Recovery

Contaminants gradually cumulated on the surface and inside the pore membrane module in the test and caused the decline in the specific flux and membrane fouling. When *SF* decreased below 10 L/(h∙m^2^∙mH_2_O), physical and chemical cleaning should be performed to remove the foulants in the membrane. In the test, the fouled membrane was cleaned, following a sequence of tap water cleaning to remove the cake layer, acid cleaning to remove the inorganic scale, and sodium hypochlorite cleaning to remove the organic compounds, such as soluble microbial products and extracellular polymeric substances.

Table 3 shows *SF* recovery after physical and chemical cleaning. After tap water washing, *SF* recovered to 25.4 L/(h∙m^2^∙mH_2_O), with a corresponding recovery of 54.1%. After acid cleaning for 0.5 and 1 h, *SF* recovered to 26.5 and 27.5 L/(h∙m^2^∙mH_2_O), indicating that inorganic scale had a little effect on the fouling. Furthermore, the membrane was cleaned using sodium hypochlorite for 3.5 h using 6 thirty-minute cleanings to remove the organic compounds attached on the surface or adsorbed in the pore. After the chemical cleaning, *SF* recovery reached 84.9%, which implied that the residual foulants could not be removed. The phenomenon could be explained by the accumulation of irreversible fouling attributed to the pore blockage caused by the foulants, which could not react with HCl and NaClO in the limited time.

## 4. Conclusions

A bench-scale membrane bioreactor was used to remove the organics (citric acid) in membrane cleaning wastewater from a thermal power plant to ensure that the effluent meets the requirements. The effects of solid retention time (SRT) and hydraulic residence time (HRT) on the removal of organic compounds, sludge performance, and membrane fouling were investigated in the test.

HRT had a great effect on the removal efficiency of organics and the membrane fouling rate. A time of 4 h was the inferior limit of HRT to ensure the MBR operates normally. The shorter SRT (15 d), the stronger the sludge activity. However, increasing the SRT had little effect on sludge performance and membrane fouling. The optimal HRT and SRT were 4 h and 30 d, respectively, to maintain the high removal of organic compounds and reduce the fouling rate and excess sludge. Relying on the strong membrane filtration, the pH values and turbidity of the effluent were quite stable. All turbidities were less than 0.7 NTU, and more than 87% of the turbidity was less than 0.4 NTU.

The membrane fouling was mainly caused by the organic compounds in the membrane wastewater. The optimal cleaning method was as follows: soaked in tap water for 2 h, in 0.05 wt% HCl for 1 h, and in 3000 mg/L for 3 h; the membrane flux could be recovered to 84.9% compared to the new membrane.

## Figures and Tables

**Figure 1 membranes-12-00755-f001:**
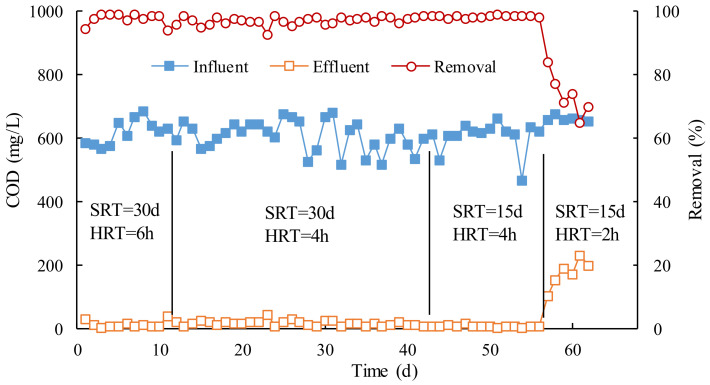
Effect of SRT and HRT on removal of organic compounds.

**Figure 2 membranes-12-00755-f002:**
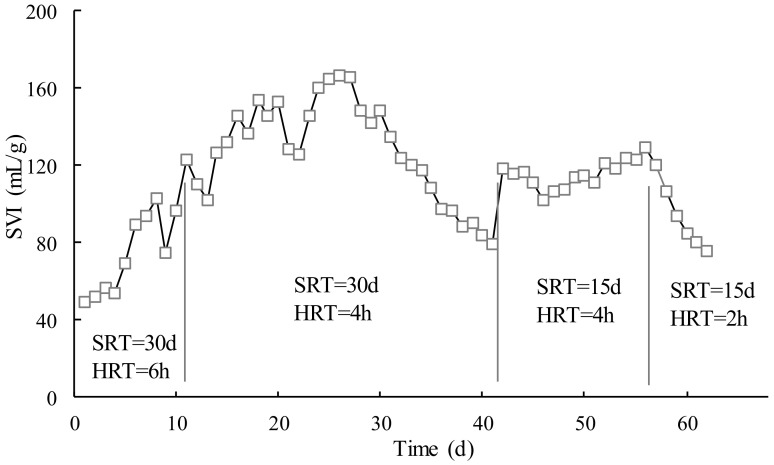
Effect of SRT and HRT on sludge performance.

**Figure 3 membranes-12-00755-f003:**
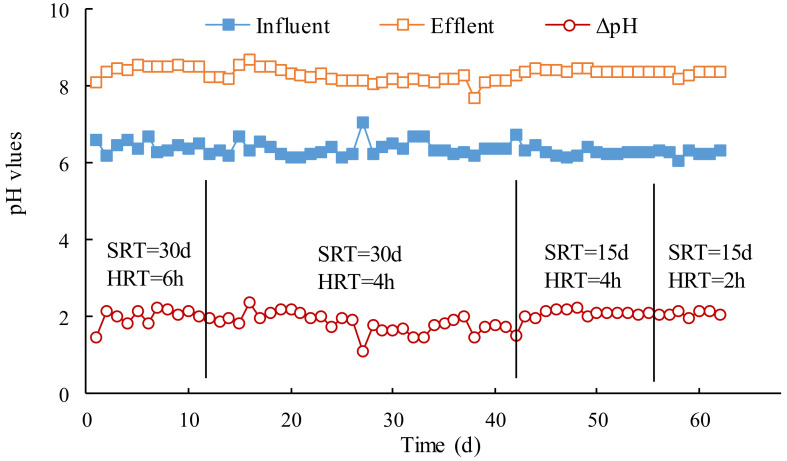
Effect of SRT and HRT on variation of pH values.

**Figure 4 membranes-12-00755-f004:**
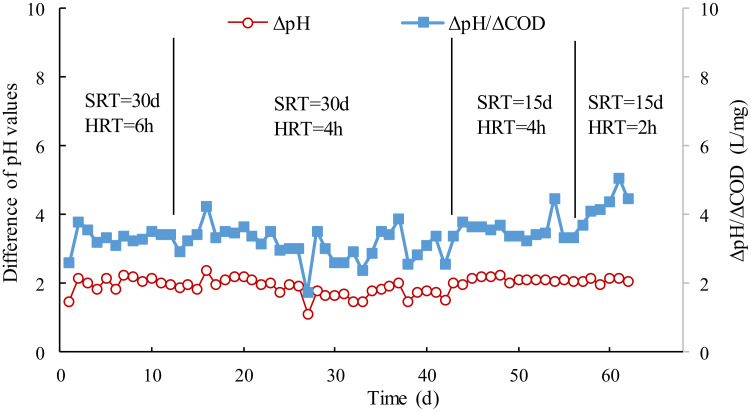
Effect of SRT and HRT on variation of pH values per unit COD (ΔpH/ΔCOD).

**Figure 5 membranes-12-00755-f005:**
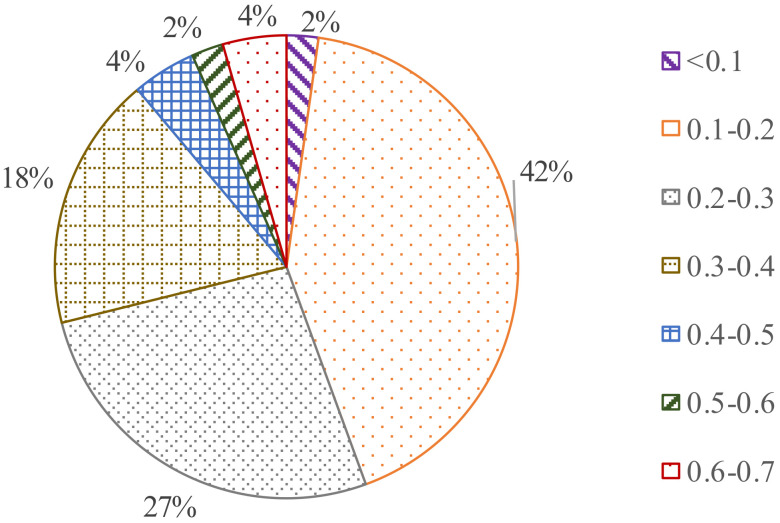
Frequency distribution of turbidity in effluent.

**Figure 6 membranes-12-00755-f006:**
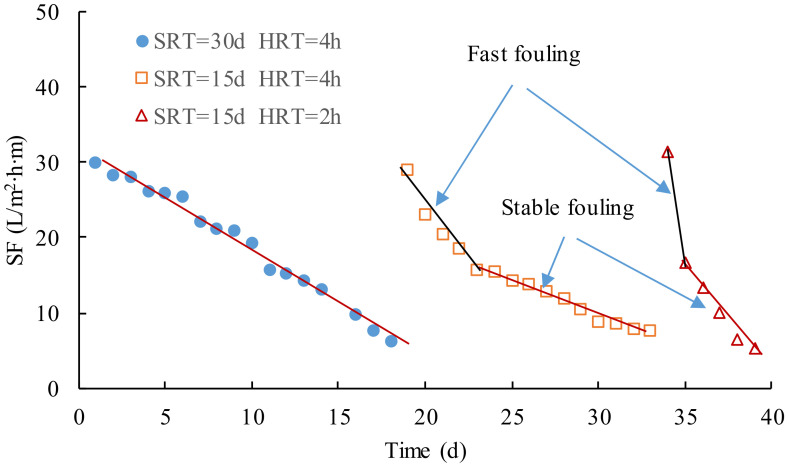
Variation of specific flux in operation.

**Table 1 membranes-12-00755-t001:** Parameters of membrane module.

Item	Parameter
Type	Hollow fiber
Material	Polyvinylidene fluoride
Nominal pore size	0.22 μm
Area	0.5 m^2^
Inner/outer diameter	0.6/1.0 mm

**Table 2 membranes-12-00755-t002:** Different fouling rates in each stage.

Stage	Fouling Rate (10^−5^/h^2^)		
	Gross	Fast	Stable
Ⅰ	5.8	5.8	5.8
Ⅱ	6.0	14.5	3.6
Ⅲ	21.7	30.4	11.2

**Table 3 membranes-12-00755-t003:** *SF* recovery for physical and chemical cleaning.

Cleaning Agent	*SF * (L/(h·m^2^·mH_2_O))	*SF* Recovery ^1^ (%)	Stage Increase in *SF* Recovery (%) ^2^
Tap water	25.4	54.1	/
HCl (30 min)	26.5	56.6	2.5
HCl (30 min)	27.5	58.5	1.9
NaClO (30 min)	31.7	67.6	9.1
NaClO (30 min)	34.8	74.2	6.6
NaClO (30 min)	37.4	79.7	5.2
NaClO (30 min)	38.8	82.6	2.9
NaClO (30 min)	39.42	84.0	1.4
NaClO (30 min)	39.86	84.9	0.9

^1^ Recovery is the SF percentage ratio of the current to the initial. ^2^ Stage increase in SF recovery was equal to the difference of the neighbor SF recovery.

## Data Availability

Not applicable.

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
