# Peer review of "Treatment of Membrane Cleaning Wastewater from Thermal Power Plant Using Membrane Bioreactor"

_membranes, 2022, doi:10.3390/membranes12080755_

Round 1

Reviewer 1 Report

Thank you for inviting me to review the manuscript titled “Treatment of Membrane Cleaning Wastewater from Thermal Power Plant Using Membrane Bioreactor”. The research aim is to investigate the treatment of membrane cleaning wastewater produced from the reverse osmosis system containing organic compounds. The study found that the optimal HRT and SRT were recommended as 18 4 h and 30 d, respectively, to ensure the removals of organic compounds, and reduce the fouling rate and excess sludge.

1-    To study the effect of HRT or SRT on the treatment performance, these parameters should be investigated on the figure x-axis (instead of Time (d)).

2-    What about the removal of several organic and inorganic pollutants?!

3-    Do not repeat the words in the title to the “keywords”.

4-    The last paragraph of Introduction should include the study objectives/procedures in brief.

5-    The study should unveil the major gaps within the existing knowledge of the proposed membrane treatment system.

6-    The results should be enriched with statistical explanations.

7-    The abstract and conclusion sections should be improved to show the main research findings.

Author Response

  • To study the effect of HRT or SRT on the treatment performance, these parameters should be investigated on the figure x-axis (instead of Time (d)).
  • Reply: I can’t understand the first point, and my explanation is to use the entire experimental process as the X-axis and HRT&SRT as parameters to illustrate in the figure.
  • What about the removal of several organic and inorganic pollutants?!
  • Reply: I’m sorry that we also have no data to support it, because the purpose of the experiment is to provide data support for industrial application and the removed pollutant was determined by actual wastewater which was mentioned in M&M in the paper.
  • Do not repeat the words in the title to the “keywords”.
  • I will revise it.
  • The last paragraph of Introduction should include the study objectives/procedures in brief.
  • I will revise it.
  • The study should unveil the major gaps within the existing knowledge of the proposed membrane treatment system.
  • I will revise it.
  • The results should be enriched with statistical explanations.
  • I will revise it.
  • The abstract and conclusion sections should be improved to show the main research findings.
  1. I will revise it.

Reviewer 2 Report

Membranes- MDPI

Treatment of Membrane Cleaning Wastewater from Thermal Power Plant Using Membrane Bioreactor

A bench scale MBR was used to treat wastewater produced from the RO unit. The impact of different operation parameters on the removal of organic compounds, the sludge performance and membrane fouling were studied.

The manuscript is discussing an important issue. The topic fits well to the scope of Membranes.

However, there are some comments and suggestions for the authors to consider before publication.

·       The introduction is short. Please, provide more information and include all relevant references.

·       The authors covered the effect of SRT & HRT on the MBR performance, however, no information about OLR (organic loading rate) and MLSS (mixed liquor suspended solids).

·       No information about the influence of temperature (e.g. : psychrophilic, mesophilic, or thermophilic) on the MBR performance.

·       The primary disadvantage of MBR is the higher capital and operating costs, kindly, cover this point ( if possible).

·       The authors measure COD only, what about BOD, TSS,TN …etc.

·       Please, rewrite cleaning procedure (make it more formal).

·       Strongly recommend to add SEM images for the dirty and cleaned membrane.

·       What is the concentration of the chemicals that you used ( HCl,  NaClO, NaOH, etc…)

·       Kindly, Rewrite the analytical method section with sufficient information.

·       What is the reason for choosing SRT= 15D, 30D & HRT= 2h, 4h and 6h? make it clear to the reader

·       I have noticed that, there is NO references from Membranes- MDPI . Please add at least three references.

Author Response

The introduction is short. Please, provide more information and include all relevant references.

  • Reply: I will revise it by provide more reference.

The authors covered the effect of SRT & HRT on the MBR performance, however, no information about OLR (organic loading rate) and MLSS (mixed liquor suspended solids).

  • Reply: Sorry, I will add the data of MLSS. The concentration of organics was always about 620mg/L as the wastewater treated was simulated wastewater of Thermal Power Plant.

No information about the influence of temperature (e.g. : psychrophilic, mesophilic, or thermophilic) on the MBR performance.

  • Reply: The MBR was aerobic by sustainable aeration treatment, so its main function was to remove COD. Moreover, the main components of actual RO membrane Cleaning wastewater we had analyzed was COD. So, in order to remove carbon source, we set water quality condition of the simulated wastewater to COD:TN:TP=100:5:1. Our emphasized goal is to remove high concentration organics by MBR and achieve a target of less than 30 mg/L of COD in effluent.

The primary disadvantage of MBR is the higher capital and operating costs, kindly, cover this point ( if possible).

  • Reply: I will try it.

The authors measure COD only, what about BOD, TSS,TN …etc.

  • We measured COD and turbidity and 87% turbidity was less than 0.4 NTU, so TSS wasn’t necessary. We prepare simulated wastewater by citric acid, so BOD didn’t make sense. We only added the nitrogen source that the microorganisms needed, which was not the main removing target.

Please, rewrite cleaning procedure (make it more formal).

  • I will revise it.

Strongly recommend to add SEM images for the dirty and cleaned membrane.

  • Sorry, the experiment was over and we didn’t conduct any characterization tests.

What is the concentration of the chemicals that you used ( HCl,  NaClO, NaOH, etc…)

  • I will add these to M&M.

Kindly, Rewrite the analytical method section with sufficient information.

  • I will revise it.

What is the reason for choosing SRT= 15D, 30D & HRT= 2h, 4h and 6h? make it clear to the reader

  • SRT and HRT were determined according to the operating parameters of the wastewater treatment plant, and the purpose of this paper is also expected to provide a reference for practical applications.

I have noticed that, there is NO references from Membranes- MDPI . Please add at least three references.

  • I will revise it.

Round 2

Reviewer 1 Report

The manuscript titled “Treatment of Membrane Cleaning Wastewater from Thermal Power Plant Using Membrane Bioreactor” has been improved based on the reviewers’’ comments, and it could be accepted for publication.

Reviewer 2 Report

The authors have addressed all comments. I am satisfied with the improvements.